# The respiratory cycle modulates distinct dynamics of affective and perceptual decision-making

Malthe Brændholt[1]*, Niia Nikolova[1], Melina Vejlø[1], Leah Banellis[1], Francesca Fardo[1,2], Daniel S. Kluger[3,4], Micah Allen[1,5]

1 Center of Functionally Integrative Neuroscience, Aarhus University, Aarhus, Denmark, 2 Danish Pain Research Center, Aarhus University Hospital, Aarhus, Denmark, 3 Institute for Biomagnetism and Biosignal Analysis, University of Muenster, Muenster, Germany, 4 Otto Creutzfeldt Center for Cognitive and Behavioral Neuroscience, University of Muenster, Muenster, Germany, 5 Cambridge Psychiatry, Cambridge University, Cambridge, United Kingdom

* malthe@cfin.au.dk

## Abstract

Breathing plays a critical role not only in homeostatic survival, but also in modulating other non-interoceptive perceptual and affective processes. Recent evidence from both human and rodent models indicates that neural and behavioural oscillations are influenced by respiratory state as breathing cycles from inspiration to expiration. To explore the mechanisms behind these effects, we carried out a psychophysical experiment where 41 participants categorised dot motion and facial emotion stimuli in a standardised discrimination task. When comparing behaviour across respiratory states, we found that inspiration accelerated responses in both domains. We applied a hierarchical evidence accumulation model to determine which aspects of the latent decision process best explained this acceleration. Computational modelling showed that inspiration reduced evidential decision boundaries, such that participants prioritised speed over accuracy in the motion task. In contrast, inspiration shifted the starting point of affective evidence accumulation, inducing a bias towards categorising facial expressions as more positive. These findings provide a novel computational account of how breathing modulates distinct aspects of perceptual and affective decision-dynamics.

## Author summary

Breathing is more than just a vital process for survival — it influences how we perceive and interact with the world around us. Recent research suggests that the rhythm of breathing, from inhaling to exhaling, affects both brain activity and behaviour. To explore this connection, we conducted a study with 41 participants, asking them to perform tasks involving visual motion and emotional facial

**Data availability statement:** All analysis code and data is available from Git repository: https://github.com/embodied-computation-group/tidal-computation/tree/main

**Funding:** o MA, MB, LB, NN and MV were supported by the Lundbeck Foundation (lundbeckfonden.com, grant number R272–2017-4345) and by the European Research Council (erc.europa.eu, grant number ERC-2020-StG-948788). FF was funded by the European Research Council (erc.europa.eu, grant number ERC-2020-StG-948838) and the Lundbeck Foundation (lundbeckfonden.com, grant number R436-2023-991). DSK was funded by the Deutsche Forschungsgemeinschaft (www.dfg.de, grant number KL 3580/1-1) and Innovative Medizinische Forschung (www.medizin.uni-muenster.de/imf, grant number KL 1 2 22 01). The funders had no role in study design, data collection and analysis, decision to publish, or preparation of the manuscript.

**Competing interests:** The authors have declared that no competing interests exist.

expressions. We found that inhaling not only sped up responses across tasks but also shaped how decisions were made. Using computational modelling, we uncovered two distinct effects: for visual motion, inhalation lowered the decision threshold, making participants prioritize speed over accuracy. For emotional facial expressions, inhalation shifted decision biases, leading participants to categorize faces as more positive. These findings reveal that breathing can subtly alter both the speed and the nature of decision-making, providing new insights into how bodily rhythms influence the mind.

## Introduction

The interoceptive rhythms of the body are essential for maintaining homeostasis [1,2]. While our breathing, heartbeat, and other visceral oscillations are critical for keeping us alive, less is known about how these rhythms influence our perception of the world. Indeed, until recently, breathing in the brain was thought to be largely constrained to basic processes concerning only the maintenance of respiratory drive and gas exchange [3,4]. This idea has recently been challenged by novel neurophysiological and behavioural discoveries [for reviews see, 5,6]. These studies have shown that breathing drives alterations in both neural oscillations and behaviour across numerous tasks and animal models.

The finding that perceptual decision-making depends on the rhythms of breathing span multiple cognitive and sensory domains. One recent study found that across visual, auditory, and memory tasks, reaction times differed substantially if participants responded during inspiration as compared to expiration [7]. Other studies have found that inspiration increases the speed of emotional processing [8], and that participants make more volitional movements during expiration [9]. These and many more recent reports suggest that breathing plays a fundamental role in shaping exteroceptive, affective [8,10,11], and cognitive behaviour [for reviews, see 5,6,12,13].

Theoretical proposals suggest that this modulation operates by adaptive gain control through the respiratory tuning of neural excitability, a mechanism that is critical for evidence accumulation during perceptual decision-making [5,14–17]. While some studies have demonstrated psychological and neural effects of the respiratory cycle that are broadly consistent with shifts in neural gain [17,18], to our knowledge, no study has directly examined the underlying cognitive mechanisms using computational modelling. As studies have highlighted consistent modulation of reaction time by the breathing cycle [7], we hypothesised that breath-brain coupling may alter the underlying latent variables governing perceptual and affective decision-making.

To test this hypothesis, we conducted a psychophysical experiment in which participants performed stimulus discrimination tasks in the domains of visual random dot motion and affective face perception. By modelling the influence of the inspiratory-expiratory cycle on perceptual choices with a hierarchical drift diffusion approach [19], we evaluated which specific decision variables in each modality were modulated by the breathing state. This enabled us to test whether respiratory state effects are primarily

elicited by changes in stimulus processing or response execution, and to evaluate the cognitive mechanisms underlying these putative effects. We confirmed that participants respond more quickly in both domains when responding during inspiration, and also found that this acceleration arises from the modulation of different decision variables across these domains. Our findings highlight distinct mechanisms by which the respiratory cycle modulates visual motion and affective face perception.

## Results

We tested the effects of respiratory state on visual and affective processing using a perceptual decision-making task which alternated between each stimulus domain across blocks. These two domains were selected because they represent fundamental dimensions of visual perception - basic sensory processing and emotional interpretation - allowing us to examine whether respiratory state effects generalise across different cognitive domains. Additionally, the careful control of perceptual and affective content in our stimuli enabled us to minimise potential biases and isolate the influence of respiration on decision-making processes in both modalities.

To maximise the ability to detect respiratory state effects on perceptual behaviour, all stimuli were individually thresholded using a Bayesian psychophysical approach [20]. Random dot motion (RDM) stimuli were calibrated to the individual motion coherence threshold for discriminating upwards vs downwards motion (mean coherence threshold=0.21, SD=0.14). For the affective domain, we adapted a face affect discrimination (FAD) procedure [21], which utilised face morphing to present stimuli conveying a range of emotional expressions between 100% happy and 0% angry and vice versa. This enabled us to titrate stimuli to the individual threshold for perceiving a facial expression as happy vs angry (mean face morph threshold=53% happy, SD=9). Following this staircasing procedure, participants completed a one-alternative forced choice paradigm during presentation of 320 near-threshold stimuli in each modality (see Fig 1 for overview).

We first assessed respiratory effects on choice behaviour.

As the perceptual decision process unfolds over time,different parts of this process may pertain to different respiratory states. To establish whether respiratory-behavioural coupling is primarily associated with perceptual or motor systems, we analysed trials grouped independently by respiratory state at stimulus presentation (stimulus-locked) and at the time of response (response-locked). It should be noted that these were analysed separately, in order to assess the independent contributions of respiration to stimulus processing and response modulation, as the study was not designed to directly compare these conditions.

In the visual motion domain, we found a significantly higher hit rate (HR) for responses made during expiration compared to inspiration (mean difference=2.0%, 95% Confidence Interval (CI) = [0-4%], $p=0.044$). This effect was not observed when grouping trials based on respiratory state at stimulus presentation (Bayes factor in favour of the null, BF01: 4.67). For FAD, the proportion of happy vs angry responses did not differ significantly for either stimulus or response grouping (BF01: 3.44 and 5.46, respectively). See S1 Fig and S1 Table for all statistics.

We next determined whether response speed for the two domains differed as a function of respiratory state, again using both stimulus and response locking. Here we found a predominant effect for response-locked median reaction times, such that responses were faster when they occurred during inspiration vs expiration for both RDM, mean difference=18.9 milliseconds (ms), 95% CI = [10-30 ms], $p<.001$, and for FAD, mean difference 21.2=ms, 95% CI = [10-30 ms], $p=.001$ (see Fig 2). In contrast, no effect for either domain was observed when analysing reaction times based on stimulus locking (BF01, RDM: 5.21, FAD: 3.4).

These results are in line with the descriptive statistics across subjects where RDM responses made during inspiration and expiration respectively had means of 675 ms±275, and 694 ms±274, just like response-locked FAD reaction times for inspiration and expiration were 737 ms±295, and 763 ms±309 respectively (±: SD). Stimulus-locked reaction times for inspiration and expiration respectively were 689 ms±273, and 688 ms±275, for RDM and 758 ms±306, and 753 ms ± 303 for FAD.

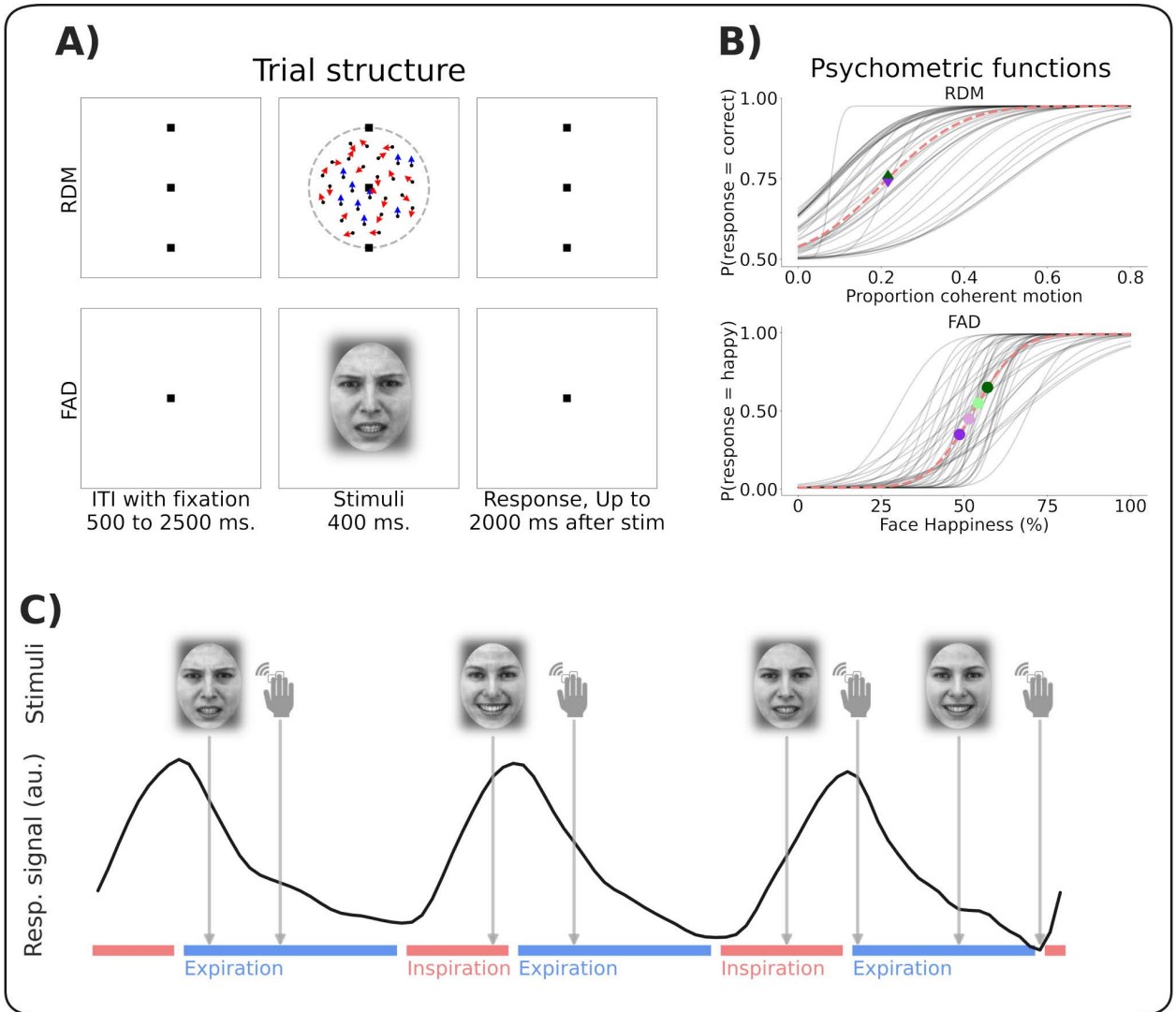

**Fig 1. A) Trial structure of the random dot motion (RDM) and face affect discrimination (FAD) task.** The task is to identify the coherent motion direction (blue arrows denote coherent motion, red arrows denote random motion. Arrows are shown here for illustrative purposes) or decided whether face expressions are happier or angrier. B) Estimated psychometric functions for both RDM and FAD stimuli. Grey lines show individual participants' traces, dashed red shows the group mean. Green and purple symbols indicate examples of the selected stimulus intensities used in the test trials. C) Each test trial was labelled both by the respiratory state at stimulus onset and at the time of response as indicated by face and hand icons, respectively. Face icons show examples of happy and angry stimuli. Red and blue lines represent periods of inspiratory and expiratory states respectively. ITI: Inter-trial interval.

In summary, responses made during inspiration were faster for both stimulus domains and less accurate only for the visual motion domain, pointing to a complex and heterogeneous coupling between respiratory state and decision-making. To establish the underlying mechanisms driving the observed coupling effects we employed drift diffusion modelling (DDM) [22]. The drift diffusion model jointly analyses choice and reaction time data. The model postulates that speeded decisions can be characterised as a noisy accumulation of evidence originating from a starting point positioned between two decision boundaries (e.g., reporting a happy or an angry face), and additional latencies not related to the decision process itself. Fitting this model to choices and reaction timesacross respiratory states allowed us to investigate

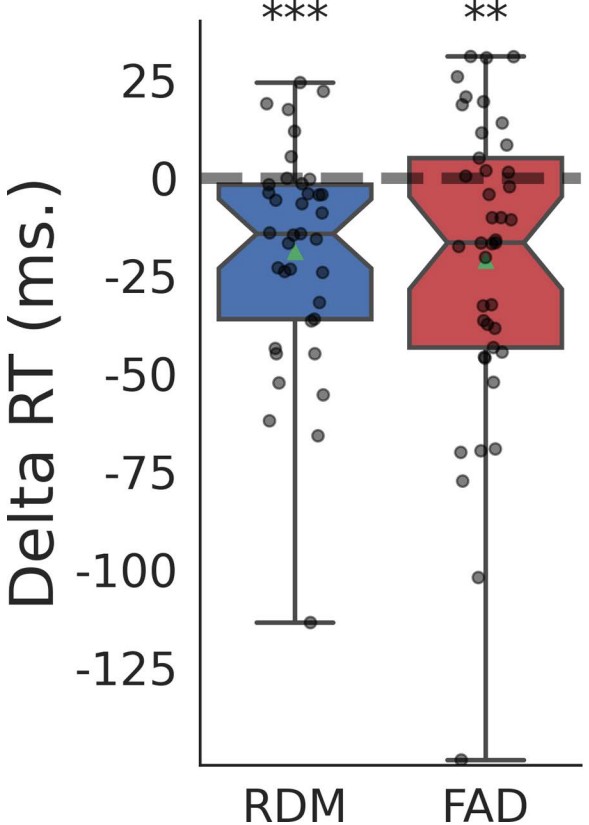

**Fig 2. Boxplots depicting the difference (delta) in median reaction time between inspiration and expiration, based on response grouping for each domain.** Positive values indicate longer reaction times during inspiration compared to expiration. Dots indicate individual participant delta medians. Notches indicate 95% CI of the median. RDM: Random dot motion, FAD: Face affect discrimination. **p < 0.01, ***p < 0.001.

whether the observed coupling effects were most effectively accounted for by changes in the rate of evidence accumulation (drift rate, *v*), participants' biases towards particular responses (starting point bias, *z*), the balance between speed and accuracy (boundary separation, *a*), or by extra-decisional variables such as attention or motor preparation (non-decision time, *t*). For a visual representation of these parameters, see Fig 3A.

We fit hierarchical DDMs where these parameters were modulated by the inspiratory vs expiratory state of respiration. Suitable model convergence and fit was assessed by calculating the Gelman-Rubin statistic (see *Methods*), and by posterior predictive checks (Fig 3B and 3C). Inspection of the group-level posterior probability distribution of each decision variable's modulation by respiratory state (as illustrated in Fig 3D) revealed a significant decrease in boundary separation during inspiration for visual motion ($P = 0.008$), consistent with observed changes in speed and accuracy described above. For the affective domain, we observed a significant shift in starting point bias towards 'happy' responses during inspiration ($P = 0.048$). For both domains, the state-related changes in evidence accumulation showed a complex, stimulus dependent pattern of respiratory modulation. That is, we found a significant reduction of drift rate for upwards motion stimuli ($P = 0.013$) during inspiration with no evidence for changes in drift rate for downwards motion stimuli ($P = 0.889$). For FAD we observed weak evidence that inspiration increased evidence accumulation for both low ambiguity stimuli (happy and angry) and for angry high ambiguity stimuli, as well as a weak trend towards a decrease in evidence accumulation for happy high ambiguity stimuli, however, none of these effects passed standard evidentiary boundaries and should thus be interpreted with caution.

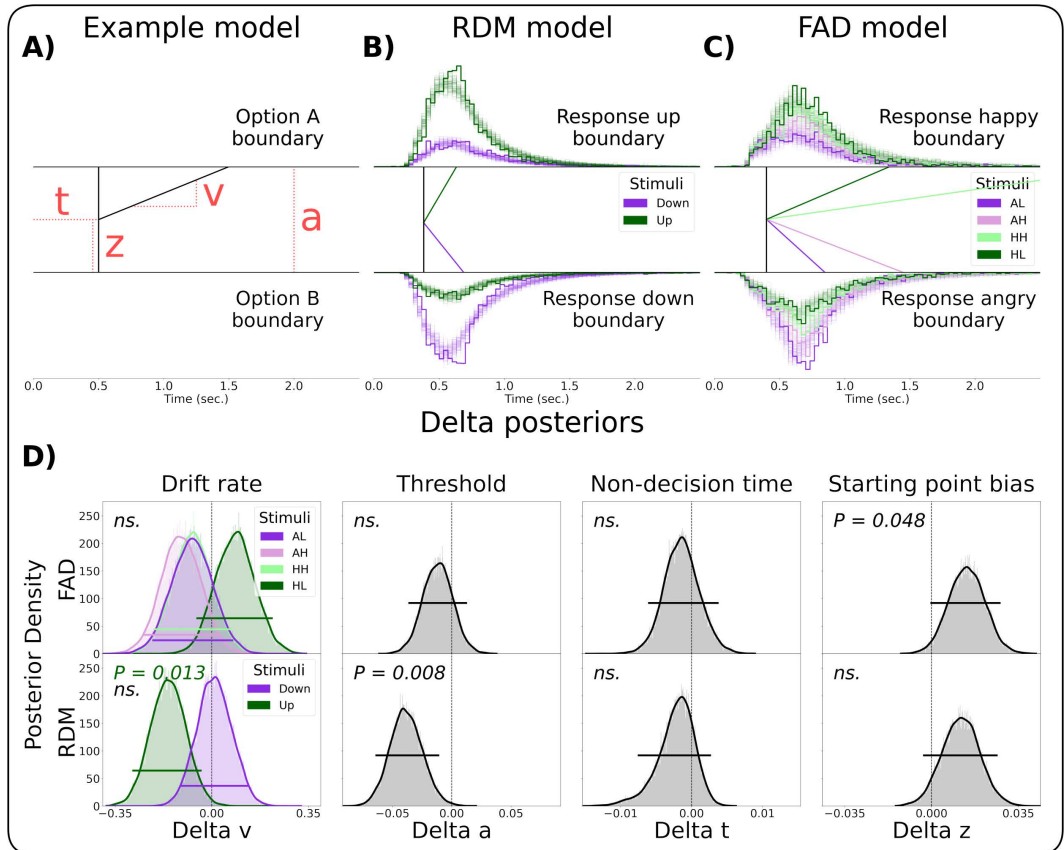

**Fig 3. Drift Diffusion Modelling of Respiratory State Effects on Decision-Making. A)** Schematic representation of the drift diffusion model (DDM) visualising the drift rate **(v)**, non-decision time **(t)**, bias **(z)**, and decision threshold **(a)**. **B)** Fitted group level DDM parameters and posterior predictive checks for the random dot motion (RDM) model and C) face affect discrimination (FAD) model. Drift rate was fitted for each stimulus class as shown by different coloured lines. Bold lined histograms represent the empirical reaction time histogram for each stimulus type and for each response option accumulated over time across all participants. Shaded lines represent the histograms of 100 data sets simulated based on the estimated model parameters. Note that the empirical and simulated data contain reaction times faster than the depicted group level non-decision time since some participants have shorter non-decision time than the group. **D)** Posterior predictive distributions for differences in DDM parameters by respiratory state. We found a significant reduction in decision threshold (P = 0.008) and drift rate of upwards stimuli (P = 0.013) for RDM, and a positive affective bias towards 'happy' responses during inspiration for FAD (P = 0.048). No significant shifts were found in drift rate of other stimuli, decision threshold in FAD or for non-decision time (all P > 0.05). Each parameter's 95% highest posterior density interval (HPDI) is shown, reflecting the effect of respiration on decision dynamics. AL: angry, low ambiguity, AH: angry, high ambiguity, HH: happy, high ambiguity, HL: happy, low ambiguity.

These effects are illustrated in Fig 3D. Additionally, we report 95% highest posterior density intervals (HPDIs) and P-values in Table 1.

## Discussion

Numerous studies have demonstrated a close linkage between the rhythms which govern brain, body, and behaviour [5,17,18,23]. While the influence of respiratory states on cognition is by now well established, so far little direct evidence has emerged to explain the computational mechanisms underlying these effects. Here, we confirmed previous findings that responses made during inspiration are associated with faster reaction times across visual and affective stimulus processing domains [7]. However, through computational modelling, we found that these surface level phenomena are explained by distinct respiratory modulation of latent decision variables. Our study sheds new light on how

**Table 1. Respiratory Delta Posteriors.**

| | Delta parameter | HPDI | P | |
|---|---|---|---|---|
| RDM | v up | [-0.288: -0.037] | 0.013 | * |
| | v down | [-0.111: 0.135] | 0.889 | |
| | a | [-0.064: -0.011] | 0.008 | * |
| | t | [-0.008: 0.003] | 0.376 | |
| | z | [-0.003: 0.027] | 0.122 | |
| FAD | V AL | [-0.216: 0.078] | 0.341 | |
| | V AH | [-0.249: 0.032] | 0.119 | |
| | V HH | [-0.211: 0.073] | 0.323 | |
| | V HL | [-0.056: 0.221] | 0.236 | |
| | a | [-0.037: 0.013] | 0.350 | |
| | t | [-0.006: 0.004] | 0.556 | |
| | z | [0.000: 0.028] | 0.048 | * |

Highest posterior density interval (HPDI) and two-sided Bayesian P-values (P) for the estimated respiratory state effects on decision making parameters. RDM: Random dot motion, FAD: Face affect discrimination, AL: angry, low ambiguity, AH: angry, high ambiguity, HH: happy, high ambiguity, HL: happy, low ambiguity. *P < 0.05.

respiratory-brain interactions control behaviour and highlights the complexity of embodied modulation of exteroceptive perception.

In the context of visual motion discrimination, we found an inspiratory reduction of decision threshold. This indicates that responding during inspiration shifts decision-making to prioritise speed over accuracy. Previous studies have indicated that respiratory state modulates cortical motor readiness [9], cortico-spinal excitability [24] and cortico-muscular communication [25], all of which point towards a possible effect of motor system activation. As such, one interpretation of our result is that the respiratory state can shift the threshold of executing actions, leading to shifts in both the frequency of spontaneous actions, as shown by Park and colleagues [9], and in the amount of evidence needed to trigger a decision response, as our findings show. This suggestion is in line with recent developments in the DDM literature that increasingly posits motor areas not merely as effectors, subject to the output of higher order decision variables, but rather as important integrators of evidence, gating the decision and thereby contributing to the setting of evidence threshold [26,27]. On this basis, we suggest that respiration may alter motor readiness during visual processing.

In the context of discriminating between angry and happy faces, we found a unimodal increase in bias towards positive emotional judgments during inspiration. This selective modulation of affective decision bias by inspiration adds further nuance to the findings of Zelano et al. [8] and Mizuhara and Nittono [11], who reported modulation of affective processing by respiratory phase. In rodents, the respiratory rhythm has also been linked to the maintenance of fear-conditioned responses, suggesting a significant role for breathing in emotional regulation [10]. Our findings highlight a distinct mechanism through which respiration may bias evidence accumulation to bring about these and other changes in affective decision-making.

Numerous authors have hypothesised that breath-brain coupling effects may depend in part on the modulation of neural excitation and perceptual tuning [17,18,23,28]. On these grounds, one might have expected to observe robust modulation of evidence accumulation across stimulus domains. In contrast, here we report a complex pattern of effects with respect to drift rate, such that evidence accumulation is selectively enhanced or suppressed in a stimulus-specific manner. For example, for visual motion discrimination, we found that inspiration is associated with a decrease in the rate of evidence accumulation only for upwards motion stimuli, whereas drift was unaffected by respiratory state for downwards stimuli. For the affective modality, no significant changes were seen for any of the stimulus types. However, we observed

trends towards increased evidence accumulation during inspiration for happy low ambiguity (HL) stimuli, as well as weak decreases in evidence accumulation for all other stimulus types. While these results should be interpreted with caution, they may indicate the possibility of stimulus specific neural tuning by the respiratory cycle across perceptual domains.

What neural mechanism could explain our findings? Breathing influences the brain through various pathways see [5] for review. Relevant examples include the brainstem mediated modulation of tonic noradrenaline [29,30], olfactory bulb propagated phase amplitude coupling [8,31–33], as well as respiratory modulation of affective and prefrontal circuits [34–37]. Other possible complementary mechanisms include respiratory coupled fluctuations in the physicochemical properties of the central nervous system, such as cardiorespiratory modulation of baroreceptors [38] and blood gas concentrations [39,40]. Given the complexity and heterogeneity of these pathways, future studies will need to directly investigate the neural mechanisms underlying respiratory modulation of decision dynamics. For example, one recent study found that the flexible modulation of decision bias depends upon alpha-band neuromodulation [41]. An intriguing avenue for future research could be to combine these approaches, for example by examining the neural correlates of respiratory shifts in affective bias, which could depend on a similar mechanism.

Our study has a few key limitations that could be addressed in future investigations. Participants viewed stimuli which were presented at peri-threshold levels. While this manipulation ensured that respiratory effects could be observed while controlling for stimulus-specific biases, it potentially introduces confounds related to drifting participant performance over time due to cognitive fatigue. While inspection of performance across blocks of trials suggested that performance was relatively stable, future studies could utilise adaptive thresholding, which would offer the further benefit of enabling estimation of respiratory-modulated psychophysics [17].

Another important limitation is that our focus on visual perception inherently restricts the generalizability of our findings. Future research could overcome this by employing varied experimental protocols and including a broader range of sensory modalities to enhance applicability across cognitive domains. While we here focus on the commonly reported effects concerning the binary treatment of the respiratory cycle, future studies could make use of more complex methods examining the link between continuous respiratory phase and behaviour.

Additionally, we did not collect eye tracking data. Prior work has shown that oculomotor behaviour, including microsaccades and saccades, is coupled to the cardiac cycle [42,43], which itself is modulated by respiration. It is therefore possible that our observed effects are partially mediated by breathing-induced changes in oculomotor rhythms. Future studies incorporating eye tracking will be needed to disentangle these contributions.

Finally, although our results suggest that breathing exerts distinct rhythms on visual and affective perception, comparable non-significant effects (e.g., bias in RDM and threshold in FAD; see Fig 3) were also observed. While we utilised a relatively large number of trials and participants for a psychophysical design, these effects might have reached significance with increased power. It may be that the overall magnitude—but not the qualitative nature—of these effects is differentially modulated. Future studies using a confirmatory design (e.g., registered reports) with enhanced within- and between-subject power could help resolve this issue.

More generally, understanding the respiratory modulation of the brain and behaviour ultimately requires a multi-modal and multi-organ perspective [See for example, 44–46]. In this study, we utilized computational modelling to explain how breathing influences choice, accuracy, and decision speed, extending and replicating previous findings. However, brain-body interactions encompass a complex array of physiological processes, including various neurotransmitters such as noradrenaline and autonomic signals from the heart, gut, and other visceral systems [43,47–49]. This complexity highlights a limitation of our current work and suggests directions for future research. Future studies could employ multimodal recordings — such as pupillometry, electrocardiograms, and electrogastrography — alongside respiratory measurements, thereby expanding computational models to account for both uni- and multimodal brain-body interactions. Indeed, recent findings on eye movements and pupillary dynamics [50], as well as animal studies demonstrating that interoceptive vagal pathways involve both multimodal and unimodal afferents [51], indicate that a comprehensive understanding of

respiratory-brain coupling necessitates a multimodal approach. Note, however, that both unidirectional anatomical connections from the preBötzinger complex to LC [30] as well as recent findings from directed functional connectivity analyses [50] suggest respiration as the primary driver of both behavioural changes and arousal markers.

Finally, while our study builds on numerous studies suggesting a causal role for breathing in modulating the brain, future work could benefit from the application of respiratory manipulations to further elucidate the directionality of breathing on specific decision variables.

## Methods

### Ethics statement

The study was conducted in accordance with the Declaration of Helsinki and approval was granted by the Region Midtjylland Ethics Committee. Formal informed verbal and written consent was obtained before participation.

### Participants

Participants were recruited from a local online participant database, Sona Systems. A total of 41 healthy human participants completed the study (mean age=26, SD=6, 27 female). Participants received financial compensation for their involvement in the study.

### Experimental setup

Participants sat upright at a desktop computer with their head stabilised by a forehead and chin rest to reduce respiratory-related head movements. The headrest and table height were tailored for comfort, setting an approximate 40 cm distance between participants' eyes and a 24-inch LED monitor with a 60 Hz refresh rate and 1920x1080 resolution. Participants used the up and down arrow keys on a mechanical keyboard to input responses.

### Physiological recordings

During the experiment, participants wore a respiration belt (Vernier, Go Direct Respiration Belt, GDX-RB), connected to the stimulus computer via a USB cable. The belt was positioned at the mid-thoracic point, aligned with the lower end of the sternum, and continuously measured force at 10 Hz. Simultaneously, a time series of experimental triggers was collected in order to align the respiratory data with stimulus and response onset.

### Stimuli

Motion perception was tested using random dot motion stimuli. All spatial dimensions are reported in degrees of visual angle. A central fixation point and two reference points at 3° over and under the fixation respectively were displayed throughout RDM trial blocks (point size=0.1°). Motion stimuli consisted of 1000 black dots on a grey background moving within a circular aperture 6° in diameter. Each dot was 1 pixel in size, moved at a speed of 9°/s and had a lifespan of 5 frames, after which they were replaced randomly within the aperture. *Signal* dots comprised the fraction of dots which displayed coherent motion upwards or downwards, whereas *noise* dots moved in a new random direction every time they were replaced (see Fig 1A for an example stimulus).

The emotional stimuli comprised 201 morphed faces that transitioned in discrete steps from fully happy to fully angry. These were generated based on the Karolinska Directed Emotional Faces (KDEF) dataset [52]. To minimise face-specific biases, we created two average anchor faces for happy and angry expressions using WebMorph [53]. Nine individual faces with the highest normed ratings for happiness or anger in the KDEF dataset were averaged for each anchor. These faces met additional criteria, such as showing teeth in both expressions, as teeth have been identified as significant valence cues [54,55]. We then generated 201 morph levels between these anchor images (See S4 Fig for examples). The

low-level properties (i.e., spatial frequency, luminance and contrast) of the images were equalised across all images using the SHINE toolbox [56]. Specifically, the Fourier amplitude spectra were matched and the mean luminance and contrast were normalised. Each image was further processed with a Gaussian kernel, causing the face to gradually blur into the background.

To ensure adequate statistical power for our within-subject analyses, we recruited 41 participants and collected 320 trials per participant. This sample size represents a 1–3 fold increase in total statistical power compared to previous studies in this field (e.g., Johannknecht and Kayser, 2022 [7]; Kluger et al., 2021 [17]; Zelano et al., 2016 [8]), ensuring adequate power to detect and model effects typical of respiratory modulation of behaviour.

## Task

For RDM stimuli, participants decided whether each stimulus was predominately upwards or downwards, with the level of motion coherence titrated to each individual's threshold. For FAD stimuli, participants viewed titrated face morphs and decided whether each stimulus was happier or angrier. These stimulus levels were calibrated to each individual's subjective perception, comprising four stimulus levels: two just above (happy, high and low ambiguity) and two just below (angry, high and low ambiguity) their affective perceptual threshold. The structure of the task consisted of a self-paced introduction to each of the two stimulus domains followed by a staircase procedure establishing thresholds for each domain (see *Staircasing procedure,* below).

After the thresholding procedure, participants completed four blocks of 80 stimuli for each domain, resulting in a total of 320 trials per domain. This number of trials was determined based on pilot studies indicating that 320 trials could be comfortably completed within a one-hour session, balancing the need for sufficient statistical power with participant comfort. In all blocks, the stimulus types were equally frequent and appeared in a pseudo-randomized order. For the RDM task, every set of 10 trials contained 5 upwards and 5 downwards motion stimuli, while for the FAD task, each set of 8 trials included 2 stimuli of each emotion level. Self-paced breaks separated the blocks to reduce fatigue. The blocks alternated between the two domains in a counterbalanced order across participants.

For both domains, the stimuli were presented for 400 ms. Responses were recorded from the stimulus onset and up to two seconds after the offset, followed by a variable inter-trial interval (ITI). To avoid spontaneous alignment of participants' respiratory cycle to the frequency of stimuli presentation the ITI was jittered according to a uniform distribution ranging from 500 to 2500 ms (see Fig 1A). The experiment and stimuli were implemented using PsychoPy [57].

## Stimulus threshold estimation

To ascertain participants' perceptual thresholds for both domains, we employed a Bayesian staircasing procedure known as the 'Psi' method [20]. This technique adaptively estimates the individual psychometric function, which represents the probability function correlating stimulus intensity with a specific binary outcome (see Fig 1B). Each staircase comprised 50 trials for both tasks, and in the event of missed responses, the trials were repeated with the same stimulus intensity.

For the RDM stimuli the threshold was defined as the coherence level at which participants had a 75% probability of providing a correct response. Psi parameters were as follows: intensity range: 0–1, threshold range: 0-0.5, slope range: 0.01-0.2, intensity precision: 0.01, threshold precision: 0.01, slope precision: 0.1, guess rate: 0.05, step type: 'lin', expected minimum: 0.5. The mean estimated coherence threshold across participants was 0.21, SD = 0.14. We then used the estimated psychometric functions to present dot-motion stimuli calibrated to each participant threshold in the RDM test-trials, resulting in an observed mean hit rate of 74% across participants.

For the FAD stimuli the threshold was defined as the point of subjective equality (PSE) of the emotional stimuli, corresponding to a 50% probability of responding 'happy.' The Psi parameters were as follows: intensity range: 0–200, threshold range: 0–200 (i.e., corresponding to 0–100% happy in 0.5% steps), slope range: 0.001-50, intensity precision: 1,

threshold precision: 1, slope precision: 1, guess rate: 0.02, step type: 'lin', expected minimum: 0. The mean PSE across participants was 53% happy, SD = 9. To ensure that participants did not view the same face stimulus repeatedly on test-trials, and thereby introduce variability across trials, we applied the estimated psychometric functions to identify four emotion stimuli centred around each participant's PSE, corresponding to a probability of categorising a stimulus as 'happy' at 35% (angry, low ambiguity), 45% (angry, high ambiguity), 55% (happy, high ambiguity), and 65% (happy low ambiguity) respectively. These stimulus levels were then presented during the FAD test-trials. This manipulation successfully manipulated affective perception, resulting in mean proportions of 'happy' responses at 32%, 42%, 53%, and 60% across the four stimulus levels.

## Analysis

### Behavioural and physiological preprocessing

Missed trials and trials with a reaction time < 100 ms were excluded from the analysis. Further, participants with extremely biased responses, indicated by an absolute criterion > 0.6 in the RDM task, were excluded from RDM-related analyses. Criterion was calculated by averaging the inverse of the cumulative normal distribution for the proportion of correct responses to upwards-motion stimuli and the proportion of incorrect responses to downwards-motion stimuli and multiplying the result by negative one [58].

The respiratory signal was low-pass filtered using a 5th order digital Butterworth filter with a cut-off frequency of 1 Hz (using the python scipy.signal.butter function), then standardised as z-scores. After preprocessing, each respiratory trace was visually inspected and segments with movement artefacts annotated. To extract the respiratory state from the preprocessed respiratory signal, peaks were identified using a peak detection algorithm (scipy.signal.find_peaks in python with the parameters prominence: 0.2, distance: 10, width: 2). The minimum values between peaks were identified and labelled as troughs. See Supplementary online material (https://github.com/embodied-computation-group/tidal-computation/tree/main) for peak and trough detection and manual annotation for each participant.

The entire respiratory timeseries was then binarised into inspiratory segments, from trough to peak, and expiratory segments, from peak to trough. Stimulus onsets and responses which occurred exactly at the peak or trough were defined as relating to respiratory transitions and were not included in the respiratory state analyses, as these have been shown to represent distinct cognitive [59] and excitatory phases [18], and there were too few in total (approx. 5% of total trials) to justify separate transition-based analyses (see Fig 1C).

Trials were then analysed separately grouped by respiratory state at stimulus onset and response, respectively. After behavioural and respiratory preprocessing a total of 37 RDM and 41 FAD datasets were available for analysis. For the included participants, a mean fraction of 7.6%, SD = 2.6, of trials were excluded for RDM in stimulus grouping and 7.4%, SD = 3.0, for response grouping. For FAD 7.9%, SD = 3.3, of trials were excluded in the onset based analysis and 8.0%, SD = 3.5, in the response based analysis. See in S2 Table for details on the exclusion steps.

### Paired T-tests

For RDM the hit rate (proportion correct responses), and for FAD the proportion of 'happy' responses, were calculated for each participant in each respiratory state for each of the two trial groupings. Similarly, the median reaction time was extracted for each of these trial groupings.

To test for an effect of the respiratory state on perceptual behaviour we employed paired t-tests for each of the outcome measures, comparing inspiration to expiration for stimulus and response grouping respectively. To evaluate the strength of evidence for any null effects, null Bayes Factors were calculated for each statistical comparison, using the default priors in the Pingouin analysis package (see S1 Fig and S1 Table).

All statistical analyses were conducted in Python using the Pingouin package v. 0.3.12 [60].

PLOS Computational Biology

## Computational modelling of respiratory state effects

We used the HDDM python package, v. 0.9.7 [19], to fit hierarchical Bayesian DDMs. In the Bayesian framework, parameters are estimated as posterior probability distributions given the data and the priors. As, to our knowledge, no DDM of respiratory effects on perceptual behaviour has been performed so far, we used an uninformative, flat prior distribution for all parameters. The hierarchical nature of the model means that parameters are estimated both at group and participant-level. More precise participants have a greater impact on the group estimate, while simultaneously, the group estimate influences individual participant estimates, pulling them closer to the group mean. This partial pooling balances between and within-participant random effects, which improves both group [19] and participant-level parameter estimation [61].

## Model formulation

We fit choice and reaction time data from the two stimulus domains independently in a stimulus-coded response model, where drift rate is estimated for each stimulus class and where the response boundaries represent the two possible response categories ("upwards" vs "downwards" and "happy" vs "angry" respectively). For respiratory effects we added group level parameters signifying the difference in a given parameter during inspiration relative to expiration for all model parameters ($v$, $t$, $a$ and $z$).

## Model sampling

The models were estimated using Markov chain Monte Carlo sampling as implemented in the HDDM toolbox. Each model was sampled using 4 independent chains with 10,000 samples, a burn in of 2000 samples and a thin of 2. Convergence was tested by calculating the Gelman-Rubin statistic [62] (GRS), comparing within chain variance to between chain variance on all parameters and confirming that the GRS was less than 1.02 [19] for all parameters of both models and by visually inspecting the sampling traces of all group level parameters (see S2 and S3 Figs).

Further posterior predictive checks, plotting the actual reaction time distributions against simulated data based on the estimated parameters, were performed (see Fig 3B and 3C) to ensure that the model captured key elements of the observed reaction time and choice data.

## Parameter tests for effect of respiratory state

We conducted Bayesian hypothesis testing [63–67] to assess the significance of respiratory effect parameters of the models, which represent differences from inspiration to expiration. For the posterior probability distributions of these parameters, we calculated the probability mass suggesting a positive and negative effect respectively. Significance was determined based on whether two times the smallest of these probability masses (P) fell below 5%, corresponding to a two-sided test with alpha level = 0.05 [68] (see Fig 3D).

## Supporting information

**S1 Fig. T-tests.** Difference in task behaviour during inspiration vs expiration per participant. Positive values indicate higher values during inspiration compared to expiration. HR: Hit rate, RT: Reaction time, C: choice (proportion 'happy'-responses). Y-axis show difference in proportions for HR and C and difference in median RT for RT. Notches indicate 95%-CI of the median. RDM: Random dot motion, FAD: Face Affect Discrimination. *p < 0.05, **p < 0.01, ***p < 0.001.
(PDF)

**S2 Fig. Sampling traces, RDM model.** Caterpillar plots showing sampling traces for the Random Dot Motion model for all group level parameters.
(PDF)

**S3 Fig. Sampling traces, FAD model.** Caterpillar plots showing sampling traces for the Face Affect Discrimination model for all group level parameters.
(PDF)

**S4 Fig. Face morphing.** Angry and happy face stimuli were morphed to create 201 categorical stimulus levels ranging from 100% angry and 0% happy to 100% happy and 0% angry.
(PDF)

**S5 Fig. Histograms of event distribution over continuous (circular) respiratory phase.** A) Random dot motion, B) Face affect discrimination. Histograms showing the relative frequency of stimulus onsets (blue lines) and responses (red lines) over the respiratory cycle. Negative pi to zero denotes the inspiratory phase and zero to positive pi denotes the expiratory phase. The green line represents the time spent in each of the phase bins. We note two things 1: As expected more time is spent at the expiratory phase compared to the inspiratory phase. 2: Relative frequency of stimulus onsets and responses follow the time spent in each phase bin.
(PDF)

**S6 Fig. Box Plots representing distribution of hit rates per breath duration quartile bin over subjects for RDM.** A) Stimulus-locked, B) Response-locked.
(PDF)

**S7 Fig. Task performance over the respiratory cycle.** Boxplots representing behaviour in eight discrete respiratory phase bins. Stimulus- and response-locked hit rate for RDM and happy rate for FAD.
(PDF)

**S1 Table. T-tests.** Statistical analysis of behaviour during inspiration vs expiration. Diff. and 95%-CI refers to the mean difference between inspiration and expiration measured in proportions for HR and C and in seconds for RT. Positive values indicate higher values during inspiration compared to expiration. T and p-values refers to paired t-tests. BF10 and BF01 is the Bayes factor in favour of the alternative and the null hypothesis respectively. Behav.: Behavioural parameter. HR: Hit rate, RT: Reaction time, C: choice (proportion 'happy'-responses), RDM: Random dot motion, FAD: Face Affect Discrimination.
(PDF)

**S2 Table. Trial exclusion.** For each task and trial grouping mean ± SD in percentage points of excluded trials per step relative to the presented 320. Behav.: Exclusion based on behaviour, resp.: respiratory, RDM: Random dot motion, FAD: Face Affect Discrimination.
(PDF)

**S3 Table. Trial exclusion, reaction times.** Reaction times (ms) following trial exclusion steps. For each task and trial grouping mean ± SD reaction times over responses for all subjects are presented. Behav.: Exclusion based on behaviour, resp.: respiratory.
(PDF)

## Author contributions

**Conceptualization:** Malthe Brændholt, Micah Allen.

**Data curation:** Malthe Brændholt, Micah Allen.

**Formal analysis:** Malthe Brændholt.

**Funding acquisition:** Micah Allen.

**Investigation:** Melina Vejlø.

**Methodology:** Malthe Brændholt, Daniel S. Kluger, Micah Allen.

**Project administration:** Melina Vejlø.

**Software:** Malthe Brændholt, Niia Nikolova.

**Supervision:** Daniel S. Kluger, Micah Allen.

**Visualization:** Malthe Brændholt.

**Writing – original draft:** Malthe Brændholt, Micah Allen.

**Writing – review & editing:** Malthe Brændholt, Niia Nikolova, Leah Banellis, Francesca Fardo, Daniel S. Kluger, Micah Allen.

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
