## [Decision Letter · Decision Letter 0]

27 Aug 2024

Dear Mr. Brændholt,

Thank you very much for submitting your manuscript "The respiratory cycle modulates distinct dynamics of affective and perceptual decision-making" for consideration at PLOS Computational Biology. I apologize for the delay in the review process, which was caused by the difficulty in finding suitable reviewers.

As with all papers reviewed by the journal, your manuscript was reviewed by members of the editorial board and by several independent reviewers. In light of the reviews (below this email), we would like to invite the resubmission of a significantly-revised version that takes into account the reviewers' comments.

We cannot make any decision about publication until we have seen the revised manuscript and your response to the reviewers' comments. Your revised manuscript is also likely to be sent to reviewers for further evaluation.

Sincerely,

Tianming Yang

Academic Editor

PLOS Computational Biology

Andrea E. Martin

Section Editor

PLOS Computational Biology

Reviewer's Responses to Questions

**Comments to the Authors:**

Reviewer #1: The manuscript by Brændholt et al reports the co-modulation of decision-making dynamics with respiration cycle during two different discrimination tasks, one based on motion coherence and another on affect in a face. Utilizing a computational model, the paper attempts at yielding mechanistic insights and an explanation for the observed behavioral effects. The study tackles an interesting scientific issue and the authors utilize pertinent computational methods for modeling the behavioral data. The paper is nicely written and structured with a clear message. In order to further contextualize the presented results, I add some aspects which the authors may consider. These may help in shedding further light on the presented findings.

Please find suggestions/comments below:

- It is unclear if the subjects were instructed to fixate during the presentation of the motion discrimination stimuli and if their eye movements were monitored to manage their oculomotor behavior. This is important, as vertically drifting motion stimuli utilized here may elicit eye movements in the subjects. Further, it has been previously suggested that the eye movements, such as the optokinetic nystagmus elicited in response to vertical motion may be asymmetric with different gains for upwards or downward direction (see, for e.g. – van den Berg and Collewjin, Experimental brain Research, 1988,

https://pubmed.ncbi.nlm.nih.gov/3384058/). It is of note that the manuscript mentions an asymmetric reduction in drift rate for motion stimuli (Line 169- 171). A characterization of the eye movement dynamics in the experiments reported in the manuscript may help ascertain if they display an asymmetric nature. Further, the modeling results may be interpreted in the light of the findings related to the eye movement dynamics.

- Recent work suggests a respiratory pupillary phase effect (Schaefer et al, 2024, https://www.biorxiv.org/content/10.1101/2024.06.27.599713v1), that is - fluctuations in pupil diameter co-occur with inhalation and exhalation. In particular, the study found that the pupil size is smallest around inhalation onset and largest around peak exhalation. In addition, pupillary dynamics have been reported to be associated with performance during a discrimination task (e.g. Mathot and Ivanon, 2019) and also associated with other cognitive processes (Ebitz ad Moore, 2018, https://www.ncbi.nlm.nih.gov/pmc/articles/PMC6350273/ ). Therefore, the authors may consider quantifying modulation of pupillary dynamics in conjunction with respiration as subjects perform the motion discrimination or affect discrimination task. This may help characterize, if the observed effects are associated with changes in the pupil diameter, which may help understand the association of respiratory cycle with the decision making dynamics better.

- The manuscript reports that the stimuli utilized for the discrimination tasks were titrated for each individual. Therefore, one may wonder if the observed effects only occur under such constrained stimulus conditions. The authors may consider discussing this point. In general, would similar co-modulation of respiratory and decision dynamics be also observed if such a stimulus threshold is not chosen.

- A general point is related to the interpretation that the respiration cycle - modulates - the dynamics of decision making. Such interpretation and similar statements in the text may suggest a causality to the reader between the two variables. Is it plausible that a co-occurring third factor, e.g. pupil diameter, or oculomotor behavior, etc may contribute to the observed reaction time related effects. Therefore, an analysis of oculomotor behavior and pupil diameter (as suggested earlier) would be helpful to add further context to the presented results. The authors may consider discussing these points depending upon the results they get, which would help to balance the overall message presented in the manuscript.

Minor comments:

- The text on Supplementary figure 2 is not quite legible. Please consider revising it.

- In general, it may help to avoid the word ‘unique’ in the following lines - e.g. line 78, 244, 297, as this may suggest and be misunderstood as the ‘only’ mechanism.

- Adding a description of the task within the caption of figure 1 may help the reader. At the same place, a description about the motion stimulus may be added – e.g. blue arrows denotes the coherent motion direction, while red arrows denote random directions, etc.

- Perhaps it is mentioned in the manuscript and I missed this – please add the average reaction time (and the standard deviation) computed over all trials during the different discrimination paradigms before and after implementing the various exclusion criterion (reported in S5 table). Further, such a reaction time statistic may be added in the main text computed for trials grouped by respiratory state at stimulus onset and response.

Reviewer #2: The authors investigate the influence of respiratory rhythms on perceptual and affective processes beyond homeostatic survival. Recent studies in both humans and rodents suggest that neural and behavioral oscillations are modulated by the respiratory cycle. To examine these mechanisms, the authors conducted a psychophysical experiment with 41 participants who categorized dot motion and facial emotion stimuli in a standardized discrimination task. The findings revealed that inspiration accelerated responses in both domains. Using a hierarchical evidence accumulation model, it was determined that inspiration lowered evidential decision boundaries, leading participants to prioritize speed over accuracy in the motion task. Additionally, inspiration shifted the starting point of affective evidence accumulation, causing a bias towards more positive categorization of facial expressions. This study provides a novel computational understanding of how respiratory rhythms impact decision-making processes.

This is an interesting and timely manuscript. The paper significantly advances our understanding of the interplay between respiratory states and cognitive functions, it offers nice insights that could inform future studies in various fields. I recommend some minor changes, mostly to clarify concepts and better contextualize the study. Plus, I recommend another analysis: participants active coupling to the expected stimuli onset.

Introduction:

1. The authors should note why the choice of those physiological tasks (rationale for each individual task, and why both of them). Plus, note here or in the discussion that two branches have emerged in the interoception literature: one focuses on symbolic stimuli containing meaning and semantic information (e.g., images of emotional faces), while the other uses more ‘perceptual’ stimuli (e.g., sounds, neutral shapes, taps). The latter branch employs stimuli that do not easily elicit evocative depictions of information and are often used to test participants' sensitivity. The disparity of results (including the current study) seems to tap into this classification. For the former, variations in the strength of external sensory data, such as facial expressions, might bias participants' responses toward a particular interpretation, related to participants' schemas, biases, or naturally prone responses when certain attributes of the sensorium are transiently modified.

2. Along the whole paper: make clearer what is meant respiratory rhythms (brain rhythms elicited by respiration, respiratory phases of the cycle per se [as registered by the respiratory belt] or both)

3. In the intro and overall paper: clarify what is meant by modalities? (Page 5, line 78).

4. Page 8, this sentences is not clear: ‘To establish whether respiratory-behavioural coupling primarily depends on aligning perceptual or motor states with the respiratory cycle’

Results:

A prominent finding in the literature is that subjects align their respiratory cycle with expected stimuli, improving performance. For instance, Perl et al. (2019) found participants inhaled about 1.5 seconds after task onset in various tasks, and Kluger et al. (2021) observed lower perceptual thresholds in a spatial detection task shortly after inhalation. Similarly, Grund et al. (2022) found that participants aligned their respiratory cycles with anticipated stimuli, enhancing tactile sensitivity during the transition from late inhalation to early exhalation, peaking at exhalation. Here, I truly recommend the author to check whether the participants coupled their respiration to the expected onset of the stimuli. It should not be difficult to compute and this analysis may shed more light on the current data.

Discussion:

There is an emerging matter in the literature that the authors could note: breathing (voluntary) as a way to control the unstoppable heart, which also introduces perceptual changes in exteroceptive tasks. The authors here could note the work of Esra al., (2020), Galvez-Pol et al., (2020, 2022), and/ or Grund et al., (2022). This could fit also the idea of respiration as an act of active sensing.

Methods:

1. The rationale for using 320 near-threshold stimuli should be explained, particularly regarding how this number was determined.

2. Additionally, it should be clarified whether the number of trials was sufficient to account for stimuli encoded in each respiratory phase and responded to in each phase, considering that this would ideally involve approximately 90 trials spread across inspiration and exhalation for both encoding and response phases.

3. The justification for the number of participants should be provided, based on sample size estimation and comparisons with other studies in this field.

Signed. A.G-P

Reviewer #3: In this study the authors designed an experiment to study the modulation of decision time in two different perceptual tasks by the respiration cycle. This study builds on previous findings on changes to both perception and response time as a function of the cardiac cycle. The study is well designed and the results are really clear. However, I have some comments that would help the reader (including me) to understand the interpretation of the results and the modelling.

Major comments.

All the significant observed effects are when the timing of the respiration cycle is taken from the response time and there were no effects when aligned to the time of stimulus onset. I might have missed this in the analysis but why was the data not analysed with an ANOVA design with factors of stimulus onset and condition? Would this not be a better test of the effects?

Did people change the length of inspiration and expiration during the experiment? It possible to analyse the data to see if the length of the phase was modulated by HIT and MISSES trials.

Relatedly - was there a difference in responses depending on when in the inspiration phase the effects occurred. Did it matter whether the stimulus or response time occurred in the first or second half of the inspiration/expiration. It was mentioned that the stimuli will go over more than one phase but it would be good to know how this actually occurred.

One imagines that if a subject perceived the stimulus (i.e Hits) that response time would be quicker than for a miss. Was this the case and is so does that RT effect only occur for HIT/MISSES or both. In other words does the modulation in HIT rate explain the RT change?

My major concern is that if there is dependency between HITs and RT (faster RT for HITS) that this would have a major impact on the modelling as the two are analysed independently. Therefore it is really critical to show that RT is not dependent on choice for the interpretation of this data to hold.

**Have the authors made all data and (if applicable) computational code underlying the findings in their manuscript fully available?**

Reviewer #1: None

Reviewer #2: Yes

Reviewer #3: Yes

PLOS authors have the option to publish the peer review history of their article (what does this mean? ). If published, this will include your full peer review and any attached files.

**Do you want your identity to be public for this peer review?** For information about this choice, including consent withdrawal, please see our Privacy Policy .

Reviewer #1: No

Reviewer #2: No

Reviewer #3: No
---

## [Decision Letter · Decision Letter 1]

28 Feb 2025

PCOMPBIOL-D-24-00926R1

The respiratory cycle modulates distinct dynamics of affective and perceptual decision-making

PLOS Computational Biology

Dear Dr. Brændholt,

Thank you for submitting your manuscript to PLOS Computational Biology. After careful consideration, we feel that it has merit but does not fully meet PLOS Computational Biology's publication criteria as it currently stands. Therefore, we invite you to submit a revised version of the manuscript that addresses the points raised during the review process.

Please submit your revised manuscript within 30 days Apr 30 2025 11:59PM. If you will need more time than this to complete your revisions, please reply to this message or contact the journal office at ploscompbiol@plos.org. Please include the following items when submitting your revised manuscript:

We look forward to receiving your revised manuscript.

Kind regards,

Tianming Yang

Academic Editor

PLOS Computational Biology

Joseph Ayers

Section Editor

PLOS Computational Biology

**Reviewers' comments:**

Reviewer's Responses to Questions

**Comments to the Authors:**

Reviewer #1: I thank the authors for addressing my questions and concerns. The revised manuscript has improved and represents an interesting contribution to the emerging field of how respiration influences decision making dynamics.

In response to the comment about eye movements, the response letter from the authors states that Galvez-Pol et al show that the respiratory rhythms appear to drive saccades. I believe, the paper referring to is the 2020 publication – the aforementioned paper investigated the co-occurrence of cardiac dynamics and eye movements during visual search, and it is likely difficult to infer causality about respiration driving oculomotor responses from this work. This may be considered when the limitations about the current study are presented in the manuscript.

Reviewer #2: The authors have correctly replied to all my questions/concerns. The manuscript was already in a very good shape, so for me it was a matter of clarifying certain points; which the authors have done. I have no further comments.

**Have the authors made all data and (if applicable) computational code underlying the findings in their manuscript fully available?**

Reviewer #1: Yes

Reviewer #2: None

PLOS authors have the option to publish the peer review history of their article (what does this mean? ). If published, this will include your full peer review and any attached files.

**Do you want your identity to be public for this peer review?** For information about this choice, including consent withdrawal, please see our Privacy Policy .

Reviewer #1: No

Reviewer #2: **Yes: ** A.G-P

**Figure resubmission:**
---

## [Editor Report · Decision Letter 2]

22 Apr 2025

Dear Mr. Brændholt,

We are pleased to inform you that your manuscript 'The respiratory cycle modulates distinct dynamics of affective and perceptual decision-making' has been provisionally accepted for publication in PLOS Computational Biology.

Best regards,

Tianming Yang

Academic Editor

PLOS Computational Biology

Hugues Berry

Section Editor

PLOS Computational Biology

---

## [Editor Report · Acceptance letter]

PCOMPBIOL-D-24-00926R2

The respiratory cycle modulates distinct dynamics of affective and perceptual decision-making

Dear Dr Brændholt,

I am pleased to inform you that your manuscript has been formally accepted for publication in PLOS Computational Biology. Your manuscript is now with our production department and you will be notified of the publication date in due course.

With kind regards,

Zsofia Freund
